# miRNAs as Therapeutic Tools in Alzheimer’s Disease

**DOI:** 10.3390/ijms222313012

**Published:** 2021-12-01

**Authors:** Chang Youn Lee, In Soo Ryu, Jin-Hyeob Ryu, Hyun-Jeong Cho

**Affiliations:** 1BIORCHESTRA Co., Ltd., Techno4-ro 17, Daejeon 34013, Korea; cylee083@gmail.com (C.Y.L.); nsooryu@biorchestra.com (I.S.R.); 2Department of Biomedical Laboratory Science, College of Medical Science, Konyang University, 158, Gwanjeodong-ro, Daejeon 35365, Korea

**Keywords:** Alzheimer’s disease, pharmacotherapy, microRNA, amyloid-beta, tau, neuroinflammation, synaptic dysfunction

## Abstract

Alzheimer’s disease (AD), an age-dependent, progressive neurodegenerative disorder, is the most common type of dementia, accounting for 50–70% of all dementia cases. Due to the increasing incidence and corresponding socioeconomic burden of dementia, it has rapidly emerged as a challenge to public health worldwide. The characteristics of AD include the development of extracellular amyloid-beta plaques and intracellular neurofibrillary tangles, vascular changes, neuronal inflammation, and progressive brain atrophy. However, the complexity of the biology of AD has hindered progress in elucidating the underlying pathophysiological mechanisms of AD, and the development of effective treatments. MicroRNAs (miRNAs, which are endogenous, noncoding RNAs of approximately 22 nucleotides that function as posttranscriptional regulators of various genes) are attracting attention as powerful tools for studying the mechanisms of diseases, as they are involved in several biological processes and diseases, including AD. AD is a multifactorial disease, and several reports have suggested that miRNAs play an important role in the pathological processes of AD. In this review, the basic biology of miRNAs is described, and the function and physiology of miRNAs in the pathological processes of AD are highlighted. In addition, the limitations of current pharmaceutical therapies for the treatment of AD and the development of miRNA-based next-generation therapies are discussed.

## 1. Introduction

Alzheimer’s disease (AD) is an irreversible and multifactorial neurodegenerative disorder, and the main cause of dementia in the global elderly population (>65 years of age) [1,2]. Approximately 10% of the elderly population is diagnosed with AD, and the age-related prevalence of this disease doubles every five years after the age of 65 [3,4]. AD is characterized by cognitive impairment and memory loss, which eventually result in a loss of the ability to independently perform daily activities. Therefore, as the aging population continues to increase globally, the increasing incidence and socioeconomic burden of AD continue to represent tremendous challenges to public health worldwide [5].

Although the cause of AD is unclear [6], several studies have reported that age, family history, the ε4 allele of apolipoprotein E (APOE), high cholesterol, type 2 diabetes, and cardiovascular disease are associated with the development of AD [7,8,9,10]. The main histopathological features of AD are amyloid-beta (Aβ) plaques and neurofibrillary tangles (NFTs) in the neocortex, hippocampus, and other subcortical brain regions [11,12,13], and these are believed to be causative factors of AD. Accumulated amyloid-beta plaques and NFTs in the brain tissue of Alzheimer’s patients cause neuroinflammation and apoptosis, leading to brain atrophy. However, no effective pharmacotherapies for the prevention and treatment of AD have been developed [6,14,15].

The currently developed drugs prescribed for AD patients are not fundamental treatment methods; rather, their purpose is to relieve symptoms or to delay disease progression. Therefore, the development of new therapeutic agents is required.

MicroRNAs (miRNAs) are composed of approximately 18–22 nucleotides, and are typically non-coding and single-stranded RNA sequences. miRNAs bind to complementary untranslated regions (3′-UTRs) of mRNAs to regulate target genes, resulting in translational repression or degradation [16,17]. miRNAs play an essential role in normal development, and are involved in various biological functions, including development, differentiation, proliferation, and apoptosis [18,19,20,21]. Therefore, the dysregulation of miRNAs is involved in many human diseases, including AD [22,23,24]. According to previous studies, dysregulation of some key factors, such as c-Myc and P53, increased the expression of the oncogenic miR-17-92 cluster in tumors [25]. In addition, miRNAs are closely implicated in the pathogenesis of major diseases, such as heart disease [26], hypertension [27], arthritis [28], diabetes [29], and obesity [30].

Several studies have reported that the miRNA profile changes in pathological conditions [31,32,33]. The relationships between diseases and miRNAs are currently being studied, including the ability of miRNAs to regulate multiple targets, which may be a promising tool for the study of multifactorial diseases, such as AD [34]. In addition, knockdown or overexpression of miRNAs can alter gene expression; thus, due to these properties, miRNAs have strong therapeutic potential to treat diseases [35].

In this review, a brief overview of the current therapeutic methods to treat AD, and the limitations of these methods, are provided. The role of miRNAs and the current trends of miRNA-based therapeutic strategies for AD are also discussed (Figure 1).

## 2. Current Treatment for AD

### 2.1. Pharmacotherapy for Alzheimer’s Disease

Though AD is the most common neurodegenerative dementia, no effective treatments have been developed. However, treatments that can alleviate symptoms are available for patients with AD. Until recently, five dementia treatments were approved by the Food and Drug Administration (FDA), including donepezil, rivastigmine, memantine, and aducanumab (Table 1).

Donepezil, the leading compound for the treatment of AD, rapidly and reversibly inhibits acetylcholinesterase (AChE), impeding the hydrolysis of the neurotransmitter acetylcholine, resulting in increased activity of acetylcholine [36]. Donepezil was approved by the FDA in 1996, and its side effects include nausea, difficulty sleeping, abnormal heart rhythms, and seizures [37]. In 1997 and 2001, rivastigmine and galantamine, two different types of cholinesterase inhibitor, were developed and approved for AD pharmacotherapy by the FDA. Like donepezil, rivastigmine and galantamine reversibly inhibit AChE to improve the intrinsic action of acetylcholine on cholinergic receptors [38]. Memantine is used to treat moderate-to-severe AD in the United States, Canada, Europe, China, and other countries, and was approved by the FDA in 2003. Memantine is an antagonist of the N-methyl-D-aspartate glutamate receptors (NMDAR) in the central nervous system [39].

Although these drugs have been used to treat AD, the cholinesterase inhibitors and memantine only treat patients’ symptoms, and do not prevent the progression of AD. Treatment with donepezil results in slightly improved cognitive function, but does not improve length of hospital stay, treatment costs, or caregiver burdens [40]. Memantine has been reported to safely and effectively improve cognitive function in patients with severe AD [41]. However, no clinical benefits have been reported for patients with early stage AD [41].

Aducanumab is a monoclonal IgG1 antibody that inhibits accumulation of extracellular Aβ plaques in the brain [42]. Although the drug received conditional approval from the FDA in June 2021, the history of aducanumab failure in phase 3 clinical trials remains controversial [43,44]. Nevertheless, aducanumab is the first drug to directly target Aβ plaques, which have been the focus of Alzheimer’s research and drug development for decades; aducanumab is being evaluated as an important therapeutic option for the treatment of AD.

### 2.2. Limitations of Drugs for AD

Various therapies have been developed to treat AD, though they each have limitations. The biggest obstacle to drug development is the lack of understanding of the pathophysiology of AD. As a result, existing AD treatments are being used for limited purposes, such as improving cognitive function or delaying memory decline. Therefore, no significant improvements in practical benefits, such as clinical improvement, reduced treatment costs, or shortened hospital stay, have been reported for patients receiving these therapies.

In addition, as AD is a multifactorial disease by, for example, accumulation of Aβ plaques and formation of NFTs, the effects of conventional pharmacotherapies with one target are limited. Although aducanumab reduces Aβ accumulation in the brain, no significant clinical effects were reported [42,43,45]. Therefore, an innovative approach that can efficiently control each risk factor of the multifactorial disease is required for the development of AD therapies. While multi-target therapeutics are attracting attention, their synergistic effects have not been proven, and more research is necessary.

Finally, a damaged brain has limited recovery abilities. An early diagnosis and regenerative medicine perspectives are required for the treatment of AD. As AD progresses, brain atrophy due to the loss of nerve cells cannot be reversed. Therefore, early treatment is important.

## 3. Pathological Characteristics of AD and the Role of miRNAs

miRNAs have recently been reported in many studies regarding neurodegenerative diseases, such as AD [46]. In addition, many studies reported that various miRNAs are involved in regulating Aβ, Tau and neuroinflammation in AD pathology (Table 2). In 2007, it was revealed that alterations in specific miRNAs, namely, miR-9, miR-125b, and miR-128, had been induced by the production of reactive oxygen species, likely generating neurotoxic metal sulfates in the hippocampi of patients with AD [47]. Subsequent studies have reported significant variations or decreased expression of miRNAs in different areas of the brain in patients with AD. The expressions of miR-106b, miR-29a/b/c, and miR-107 were decreased in the hippocampi of patients with AD, and miR-181c and miR-101 were deregulated in the cortices of patients with AD [48,49]. Several miRNAs, such as miR-212, are expressed at low levels in the white matter of the brain of patients with AD, and the expression of miR-15 and miR-107 is decreased in the grey matter [50]. In contrast, miR-125b, miR-128, and miR-146a have increased expression in the brains of these patients [51]. As such, several reports have revealed that miRNA is closely related to AD. In this review, we introduce the relationship between miRNAs and major factors involved in AD pathogenesis.

### 3.1. miRNAs Regulate Aβ

The amyloid hypothesis was first presented by John Hardy and David Allsop in 1991 [52]. Aβ is a transmembrane protein generated by the hydrolysis amyloid precursor protein (APP) via the amyloidogenic pathway. C-terminal fragments of APP are produced via hydrolysis by α-, β-, γ-, and η-secretase through three pathways [53]. Among them, Aβ_1-42_ is produced by the cleavage of APP by β-secretase and γ-secretase. Aβ deposited in the hippocampus and basal segment in the form of neurotoxic Aβ plaques recruits more Aβ, forming insoluble aggregates and inducing mitochondrial damage [54], disrupting homeostasis [55], and causing synaptic dysfunction [56].

Several studies have reported that various miRNAs (miR-20a, miR-106a, miR-106b, miR-17-5p, miR-16, miR-101, miR-147, miR-153, and miR-520c) directly regulate APP expression by targeting the 3′-UTR of APP mRNA [57,58,59,60]. Among these miRNAs, miR-106b, and miR-101 are downregulated in the brains of patients with AD, resulting in increased APP expression and Aβ production. Furthermore, AD-specific polymorphisms have been reported to influence the APP-modulating activity of miR-147 and miR-20a [61].

β-secretase cleaving enzyme 1 (BACE1) levels have been reported to be increased in patients with AD, increasing the risk of sporadic AD [62]. Several studies have reported a relationship between abnormal miRNA expression and increased levels of BACE1. miR-107, miR-298, miR-328, miR-15a, miR-15b, miR-195, miR-103, and miR-485 target the 3′-UTR of BACE1 [63,64,65,66,67]. These miRNAs are decreased in patients with AD [68,69,70,71]. In addition, significant reductions in miR-29a and miR-29b levels were reported to lead to an increased level of BACE1. Thus, miR-29 has been proposed as an endogenous BACE1 regulator [48,72]. These findings suggest that the miRNAs play important roles in regulating the amyloidogenic pathways in the brain of AD patients.

miRNAs can also affect AD pathology in other ways. Decreased monocyte lysosomal hydrolases, including the cathepsin B, D, and S enzymes, also play a role in Aβ accumulation [73]. Recent studies have shown that the upregulation of miR-128 is responsible for the downregulation of cathepsin B, D, and S. Furthermore, miR-128 inhibition in monocytes isolated from AD patients improved both lysosomal enzyme expression levels and Aβ degradation capacity [74]. These findings indicate that miRNAs are involved in the clearance of Aβ via the regulation of enzymatic activities in cleavage and degradation processes. Taken together, these findings suggest that miRNAs have the potential to be used as therapeutic agents or as mediators for regulation of Aβ in AD patients.

### 3.2. miRNAs Regulate Tau

Tau, a microtubule-associated protein generated by the alternative splicing of the MAPT gene, is mainly found in neuronal axons in the brain, where it maintains microtubule structure, cytoplasmic transport processes, and synaptic structure and function, and regulates neuronal signaling [75,76]. Tau is a phosphoprotein, and its phosphorylation is regulated by balanced protein kinase and protein phosphatase activities, and changes according to the stage of brain development. Under normal conditions, tau has few phosphorylation sites, and tau phosphorylation negatively regulates the binding of tau to microtubules. However, under pathological conditions, tau is hyperphosphorylated. Hyperphosphorylation of tau in the AD patient brain causes changes in microtubule configuration and the loss of tubulin polymerization, eventually resulting in defective microtubule function [77,78]. The elevated levels of cytosolic tau in AD lead to the formation of insoluble, paired helical filaments and straight filaments through tau–tau interactions and polymerization, resulting in the generation of intraneuronal fibrillar deposits [79]. These intraneuronal fibrillar deposits reduce the number of synapses, induce neurotoxicity, and eventually cause cell dysfunction [80,81,82].

Various studies reveled that miRNAs are closely involved in the regulation of tau protein homeostasis. Carrettiero et al. reported that miR-128a regulates the production of the auxiliary chaperone protein BAG2, which plays a role in the degradation and aggregation of tau [83]. Furthermore, miR-124, miR-132, and miR-9 can alter the accumulation of endogenous tau [84,85]. Both miR-34a and miR-26b can inhibit tau expression and affect NFT formation [86]. It is suggested that miR-9 expression increases concurrently with the decreased expression of sirtuin 1 (SIRT1), which participates in tau pathology as a deacetylase [85]. These findings suggest that miRNAs play crucial roles in maintaining microtubule structure and synaptic structure via regulation of tau protein homeostasis.

Phosphorylation of tau is very important in the progression of AD. The downregulation of miR-101 expression increases the levels of phosphorylated tau [87]. miR-922 binds the 3′-UTR of ubiquitin carboxyl-terminal hydrolase isozyme L1 (UCHL1), leading to a reduction in UCHL1 expression, which reduces the level of phosphorylated tau [88]. Wang et al. reported that miR-138 was increased in AD models, such as N2a/APP and HEK293/tau cell lines. In addition, miR-138 promotes tau phosphorylation by targeting the retinoic acid receptor alpha/GSK-3β pathway [89]. He et al. reported that miR-326 decreased tau phosphorylation. These findings suggest that miRNAs are involved in regulation of tau phosphorylation, which is closely related to tau pathology, such as defective microtubule configuration in AD patients. Therefore, it could be suggested that miRNAs are potential treatments that can achieve the suppression of tau-related pathological properties.

### 3.3. miRNAs Regulate Neuroinflammation

Several studies have suggested that AD is strongly related to immune mechanisms [90,91,92]. Protein accumulation, associated with the pathology of AD, stimulates receptors on microglia and astrocytes to trigger immune responses, including the release of inflammatory mediators [93,94,95]. A short-term inflammatory response removes the causative agent of inflammation and contributes to the recovery of the affected area, and a continuous inflammatory response causes irreversible tissue damage. Inflammatory responses in patients with AD are continuous, causing neuronal loss, promoting the progression of AD disease, or playing a significant role in exacerbations of AD.

Several studies have reported that the NLRP3-inflammasome is activated in microglia, increasing the expressions of inflammasome-forming factors, such as NLRP3, caspase-1, and ASC, leading to increased secretion of the pro-inflammatory cytokines interleukin (IL)-1 beta and IL-18, inducing neuronal death in patients with AD [95,96,97].

Zhou et al. reported that miR-7 inhibits the NLRP3-inflammasome activity of microglia, and anti-miR-7 activates the inflammasome. In addition, the inflammatory response was reduced by miR-7 in mouse models [98]. Cao et al. reported that the overexpression of miR-9 leads to suppression of NLRP1-inflammasome activation by regulating the expression of NLRP1 [99]. In addition, the levels of the pro-inflammatory cytokines IL-1 beta and IL-18 were also reduced. Thus, the overexpression of miR-9 may be an effective method to characterize brain damage after an ischemic stroke.

In the setting of neuroinflammation, activation of the inflammasome leads to pyroptotic cell death by cleaving the N-terminus of gasdermin D (GSDMD) and by the secretion of proinflammatory cytokines [100,101]. miR-22 targets GSDMD [102], and the expression of miR-22 was lower in patients with AD than in healthy adults. The overexpression of miR-22 in an APP/PS1 mouse model significantly improved memory and behavior, and suppressed the expression of proinflammatory cytokines, such as IL-1 beta and IL-18, by inhibiting GSDMD. This resulted in the prevention of pyroptotic cell death. Based on these findings, it could be suggested that miRNAs play an important role in controlling neuroinflammation; thus, miRNA could be an option for the treatment of degenerative brain diseases, such as AD.

### 3.4. miRNAs Regulate Synaptic Function

Increasing evidence suggests that miRNAs are involved in early synaptic alterations and synapse loss in patients with AD and in AD experimental models. For instance, miR-26b, miR-132, and miR-206 are able to target neurotrophic factors, such as insulin-like growth factor 1 (IGF1) and brain-derived neurotrophic factor (BDNF); these miRNAs regulate spine remodeling via dendrite maturation [103,104]. In addition, it is suggested that the dendritic miR-134 can facilitate homeostatic synaptic function in synapse formation through direct regulation of *LIMK1* and *PUM2* expression [105,106,107]. Moreover, miR-9 controls dendritic development by downregulation of the transcriptional repressor REST in primary culture and in the mouse brain [108]. These findings indicate that several miRNAs are involved in synaptic morphological changes, including dendritic formation and synapse maturation.

Several studies have reported that miRNAs also regulate the pre-synaptic function in AD [109,110,111]. For instance, miR-210 can bind its target, such as the synaptosomal-associated protein, *Snap25* mRNA. In a hippocampal neuron culture, overexpression of miR-210 decreased the number of synapses. In an animal model, intracerebroventricular injection of miR-210 induced the phenotypes of synaptic loss in the brain and of cognitive impairment [109]. Furthermore, overexpression of miR-485 reduced the abundance and protein expression of synaptic vesicle glycoprotein 2A (SV2A) mRNA in hippocampal neurons, leading to a reduction of spontaneous neurotransmitter release [110]. Especially, Aβ-induced miR-34c alteration can decrease the expression of vesicle-associated membrane protein 2 (VAMP2), a protein component of the SNARE complex, which, in turn, causes learning and memory deficits with synaptic failure in AD models [111]. Based on these findings, it could be speculated that the miRNAs may play a critical role in vesicle trafficking and neurotransmitter releases, via the regulation of physiological conditions in pre-synaptic neurons in AD models.

Since glutamatergic neurotransmission plays a key role in synaptic function and synaptic plasticity processes, including long-term potentiation (LTP) and long-term depression (LTD), numerous studies of miRNAs’ targets have been conducted on glutamate receptors and excitatory synapses in AD. It is suggested that miR-9, miR-92, miR-137, and miR-501 selectively regulate GluA1 trafficking, resulting in a reduction of the α-amino-3-hydroxy-5-methylisoxazole-4-propionic acid glutamate receptor (AMPAR) insertion in the cell surface [112,113,114,115]. Additionally, the increased miR-181a, by accumulation of Aβ, significantly decreased the GluA2 subunit in AMPAR and other plasticity-related protein expression (i.e., scaffolding proteins and post-synaptic density 95) in mouse hippocampi, which, in turn, causes memory deficits [116]. Like AMPAR, surface expression of NMDAR is also regulated by several miRNAs, such as miR-125b and miR-34a [117,118]. In hippocampal culture, overexpression of miR-125b suppressed GluN2A expression. In addition, miR-125b prolonged excitatory postsynaptic currents by a half-width, which is consistent with the relative loss of synaptic GluN2A-containing NMDAR [117]. While miR-34a targets the GluN2B subunit of NMDAR, it also regulates synaptic plasticity modulation in the working memory [118]. Based on these findings, dysregulation of glutamate receptor internalization (AMPAR and NMDAR) causes an imbalance between LTP and LTD in AD, creating a functional deficit in glutamatergic neurotransmission.

Taken together, these studies showed that several miRNAs are able to regulate structural and morphological alterations in pre- and post-synaptic neurons, and they may play key roles in the maintenance of synaptic function on learning and memory in the brain of AD patients.

**Table 2 ijms-22-13012-t002:** AD-related miRNAs and their targets.

AD Phenotype	miRNA	Target mRNA	Reference
Aβ	miR-16, miR-17, miR-20a, miR-101, miR-106a, miR-106b, miR-147, miR-153, miR-520c	APP	[57,58,59,60]
	miR-15a, miR-15b, miR-29a, miR-29b, miR-29c, miR-103, miR-107, miR-298, miR-328, miR-195, miR-485	BACE1	[63,64,65,66,67]
	miR-128	Cathepsin B, D, S	[74]
Tau	miR-128	BAG2	[83]
	miR-9	SIRT1	[85]
	miR-922, miR-181b	UCHL1	[88,119]
	miR-124	Caveolin-1	[120]
	miR-132	ITPKB	[121]
	miR-34a	Tau	[122]
	miR-26b	Rb1	[86]
Neuroinflammation	miR-7, miR-223	NLRP3	[98,123]
	miR-9	NLRP1	[99]
	miR-22	GSDMD	[102]
	let-7 family	TLR4	[124]
		IL-6	[125]
	miR-485	AKT3	[126]
Synaptic function	miR-26b, miR-206	IGF1	[103]
	miR-132, miR-206	BDNF	[104]
	miR-134	LIMK1, PUM2	[105]
	miR-9	REST	[108]
	miR-210	Snap25	[109]
	miR-485	SV2A	[110]
	miR-34c	VAMP2	[111]
	miR-92, miR-137, miR-501	GRIA1	[113,114,115]
	miR-181a	GRIA2	[116]
	miR-125b	GRIN2A	[117]
	miR-34a	GRIN2B	[118]

## 4. miRNA as Diagnostic AD Biomarker

Early diagnosis is essential for the most effective treatment of AD. AD lesions cause irreversible damage to the brain, so if diagnosis is delayed, treatment options and recovery are very limited. A recent report suggests that early diagnosis can cut the risk of developing Alzheimer’s by one third [127]. Currently, abnormal accumulation of Aβ is visualized to diagnose AD through brain imaging, using the cerebrospinal fluid (CSF) test and positron emission tomography. However, these methods are either invasive to the patient or require a very expensive financial expenditure [128]. In addition, cognitive function evaluation performed for AD diagnosis is used only as reference data, due to individual differences and low accuracy [129].

As aforementioned, several miRNAs that regulate Aβ production or tau have been reported to have increased or decreased expression in AD lesions. These miRNA profile changes are observed not only in brain tissue, but also in blood. Based on this evidence, the development of an AD diagnostic method based on miRNA may improve some of the limitations of the current diagnostic method. As reported by Bhatnagar et al., in 2014, miR-34c was found to be elevated in plasma from patients with AD. miR-34c inhibits the expression of genes involved in survival and oxidative defense pathways, such as Bcl2 and SIRT1. Therefore, miR-34c shows potential as a biomarker for screening patients with AD [130]. In addition, according to Wang’s report, as a result of analyzing plasma from 120 control subjects, 120 PD patients, and 120 AD patients, miR-103 is promising as a biomarker for AD disease. miR-103 was decreased in AD patients, compared to controls, and was negatively correlated with dementia severity [131]. According to a recent report in 2020 by Souza et al., a sample of women aged 55 years and older, carrying the ε4 allele of APOE, found a 3-fold decrease in miR-9 expression levels, compared to the control group [132].

However, single-stranded miRNAs exhibit a short half-life in biofluids, such as CSF or blood. Nevertheless, the reason why miRNA-based diagnostic methods are attracting attention is that miRNAs encapsulated in exosomes and microvesicles, or bound to proteins, are stably present in biofluids. In addition, in order to increase the reliability of miRNA as a biomarker of diseases, such as AD, it is necessary to additionally standardize the sampling method and sample analysis method of biofluids.

## 5. Limitation of mRNAs

miRNAs are powerful tools to regulate gene expression. However, there are some obstacles to overcome for a clinical application. First, the relationship between miRNA and the target gene is not always a 1:1 match. Thus, it is difficult to define a suitable target as the reason for considering off-targets [133]. Therefore, the development of a sophisticated target gene prediction algorithm is required to minimize the side effects. Second, natural variations in the expression patterns of miRNAs must be considered, such as gender [134,135] and age [136,137]. Third, single-stranded miRNAs exhibit very fast decay kinetics in certain situations, such as in biofluids [138,139]. Therefore, additional means, such as a drug delivery system capable of delivering miRNA to the diseased site in a stable state, are required. Finally, results detecting expression of miRNAs may vary, depending on the detection platform, so that a careful and critical evaluation system is required when interpreting the data [140]. Despite the aforementioned limitations, miRNAs are attractive therapeutic tools to treat disease, such as neurodegenerative disease, especially AD. Thus, we suggest that these obstacles should be overcome, in order to treat disease using miRNAs as therapeutic drugs.

## 6. Conclusions and Prospects

miRNAs are closely related to the pathophysiology of various neurodegenerative diseases, including AD, and may also be key factors in the treatment of such diseases. The pathological functions of genes that are regulated by multiple miRNAs overlap and interact, and the effects of miRNA networks are better than those of individual miRNAs. The regulatory effects of miRNAs are sequence-specific, and each miRNA can regulate multiple genes; therefore, the use of miRNAs is promising for the treatment and regulation of complicated disease networks and pathways. Thus, these miRNA characteristics may allow for effective treatment of multifactorial disease.

Although several drugs have been studied comprehensively, a safe and effective drug for the treatment of AD has yet to be identified. Therefore, the development of successful miRNA-based drugs for the treatment of AD is needed; also required are extensive research and in-depth knowledge of the precise mechanism of miRNA-target interactions and regulation of the target by miRNAs. Studies regarding miRNA-based therapies for AD are in the early stages, though extensive research regarding the treatment of AD has been conducted. Therefore, effective miRNA therapies for AD are likely to be developed in the very near future.

## Figures and Tables

**Figure 1 ijms-22-13012-f001:**
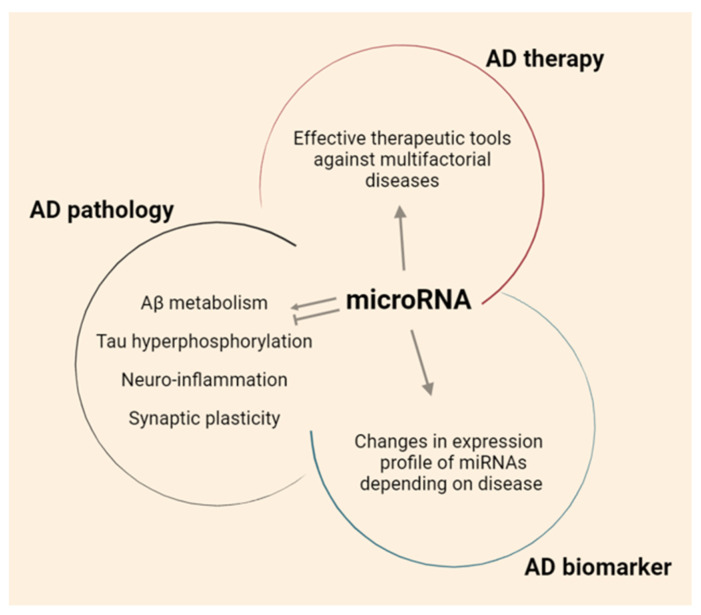
The role of miRNAs in AD.

**Table 1 ijms-22-13012-t001:** FDA-approved drugs for AD.

Drug Name	Active Ingredient	Action	Indication	FDA Approval
Aricept	Donepezil	AChE inhibition	Mild to severe AD	1996
Exelon	Rivastigmine	AChE and BuChe inhibition	Mild to moderate AD	1997
Razadyne	Galantamine	AChE inhibition Nicotinic receptor-like structure	Mild to moderate AD	2001
Ebixa	Memantine	NMDA receptor antagonists Inhibition of hyperactivity of glutamate	Moderate to severe AD	2003
Aduhelm	Aducanumab	Aβ-directed monoclonal antibody	Mild cognitive impairment or mild dementia	2021

## Data Availability

Not applicable.

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
