# Peer review of "miRNAs as Therapeutic Tools in Alzheimer’s Disease"

_ijms, 2021, doi:10.3390/ijms222313012_

Round 1
Reviewer 1 Report
The review is very well written and comprehensive.
The article entitled "miRNAs as Therapeutic Tools in Alzheimer’s Disease" describes in details the miRNAs used as biomarkers and therapeutics in AD. The authors very well described the current treatments available for AD and the limitations of these therapeutics, the different miRNAs and the target genes in AD pathogenesis e.g amyloid generation, neuroinflammation, synaptic function, Tau proteins. The authors described the future avenues associated in this field. Overall, the review is well written, timely and descriptive.
Author Response
Dear Reviewer,
We authors very much appreciated the encouraging, critical and constructive comments and suggestions on this manuscript by the reviewers.
Reviewer 2 Report
In this manuscript named “miRNA as Therapeutic Tools in Alzheimer’s Disease.” the authors mentioned the future possibilities and recent efforts in the direction of application of miRNA as a therapeutic tool to solve the multi-dimensional puzzle i.e.; Alzheimer’s Disease (AD). Initially, they described the FDA-approved drugs for AD and their short comes. Then in this context, they showed the possibilities of the miRNA to emerge as a gamechanger along with different approaches. The manuscript is well written but, there are a few suggestions as follows:
- The authors explained the effectiveness and limitations of all the FDA-approved drugs in their manuscript. It would be more interesting if authors can give an overview of what are the advantages of miRNA over these FDA-approved drugs if they have?
- Section 3 named “Pathological characteristics of AD and the role of miRNAs” requires some improvement with little more clarity. They only mentioned the observations but, the role of the miRNAs is not clear from that section.
- In section 3.4, at lines 278 and 280, the authors mentioned NR2A and NR2B of NMDA receptors. I think, nowadays, Glutamate receptors researchers use GluN2A and GluN2B instead of NR2A and NR2B (Minor thing, if the author would like to change).
- In the conclusion and perspectives, the authors ascribed only gender and age dependency as the few obstacles for the miRNA used as potential drugs. The authors mentioned that the application of miRNA will be multi-dimensional. It will be more attractive to readers if authors give more information about the obstacles and what the authors envision to overcome these obstacles.
- This manuscript is very informative as the ongoing miRNA research and FDA-approved drugs for AD were enlisted in the tables in detail. Nowadays, readers try to judge a paper in a shorter time whether that will be useful for them or not. In that case, my suggestion to the authors to put a schematic pictorial presentation of this review will be more attractive to these kinds of readers.
Author Response
Review #2
In this manuscript named “miRNA as Therapeutic Tools in Alzheimer’s Disease.” the authors mentioned the future possibilities and recent efforts in the direction of application of miRNA as a therapeutic tool to solve the multi-dimensional puzzle i.e.; Alzheimer’s Disease (AD). Initially, they described the FDA-approved drugs for AD and their short comes. Then in this context, they showed the possibilities of the miRNA to emerge as a gamechanger along with different approaches. The manuscript is well written but, there are a few suggestions as follows:
- The authors explained the effectiveness and limitations of all the FDA-approved drugs in their manuscript. It would be more interesting if authors can give an overview of what are the advantages of miRNA over these FDA-approved drugs if they have?
Response: Thank you for the comments. As mentioned, we should have written the advantage of miRNAs as therapeutic drugs compared with FDA-approved drugs. However, as you well known, there is no miRNA drug to treat AD so we suggest the capability of miRNAs as a therapeutic drugs. In addition, we mentioned roles of miRNAs involving AD, such as beta-amyloid, Tau, or neuroinflammation in Section 3. Nevertheless, there are advantages of miRNAs to use therapeutic tools to treat disease, as you well known, miRNAs are capable of multi-targeting which is probably able to overcome the limitations of conventional chemical drugs. Furthermore, multiple miRNAs are also able to regulate one target. These properties of miRNA are attractive for developing effective therapeutic drugs to treat multifactorial diseases such as AD. In addition, miRNA is a key regulator that modulates cell-biological phenomena such as cell differentiation, survival, and regeneration.
Thus, we suggested that miRNA will be promising drug to treat AD due to regulate various phenomenon.
- Section 3 named “Pathological characteristics of AD and the role of miRNAs” requires some improvement with little more clarity. They only mentioned the observations but, the role of the miRNAs is not clear from that section.
Response: We agree with the reviewer’s comments. As your recommendation, we added descriptions to clarify the roles of miRNAs in AD pathology (amyloid-beta, tau, inflammation, synaptic dysfunction) throughout Section 3 in the revised manuscript (page 5 line 161-162, 168-171; page 5 line 195-196, 204-207; page 6 line 234-237).
- In section 3.4, at lines 278 and 280, the authors mentioned NR2A and NR2B of NMDA receptors. I think, nowadays, Glutamate receptors researchers use GluN2A and GluN2B instead of NR2A and NR2B (Minor thing, if the author would like to change).
Response: As the reviewer’s suggestion, we changed ‘NR2A’ and ‘NR2B’ to ‘GluN2A’ and ‘GluN2B’, respectively, in the revised manuscript (page 7 line 273-278).
- In the conclusion and perspectives, the authors ascribed only gender and age dependency as the few obstacles for the miRNA used as potential drugs. The authors mentioned that the application of miRNA will be multi-dimensional. It will be more attractive to readers if authors give more information about the obstacles and what the authors envision to overcome these obstacles.
Response: We are very thankful to the reviewer for this insightful comment. Based on the comments of the reviewers, we have added description in section 5 (page 8 line 317-332).
- Limitation of mRNAs
miRNAs are powerful tool to regulate gene expression. However, there are some obstacles to overcome for clinical application. First, the relationship between miRNA and target gene is not always 1:1 match. Thus, it is difficult to define suitable target that’s the reason we should consider off-targets [133]. Therefore, the development of a sophisticated target gene prediction algorithm is required to minimize the side effects. Second, natural variations in the expression patterns of miRNAs must be considered such as gender [134,135] and age [136.137]. Third, single-stranded miRNAs exhibit very fast decay kinetics in certain situations, such as biofluids [138, 139]. Therefore, additional means such as a drug de-livery system capable of delivering miRNA to the diseased site in a stable state are re-quired. Finally, results detecting expression of miRNAs may vary depending on the detection platform so that careful and critical evaluation system is required when interpreting [140]. Despite the aforementioned limitations, miRNAs are attractive therapeutic tools to treat disease such as neurodegenerative disease, especially AD. Thus, we suggest that these obstacles should be overcome to treat disease using miRNAs as therapeutic drugs.
- This manuscript is very informative as the ongoing miRNA research and FDA-approved drugs for AD were enlisted in the tables in detail. Nowadays, readers try to judge a paper in a shorter time whether that will be useful for them or not. In that case, my suggestion to the authors to put a schematic pictorial presentation of this review will be more attractive to these kinds of readers.
Response: Based on the comments of the reviewers, we have added a figure. (page 2 line 69).